# Nitrate Increases Aluminum Toxicity and Accumulation in Root of Wheat

**Yan Ma [1], Caihong Bai [2,3], Xincheng Zhang [4,5,*] and Yanfeng Ding [1,*]**

1   College of Agronomy, Nanjing Agricultural University, Nanjing 210095, China
2   College of Agronomy, Yulin Normal University, Yulin 537000, China
3   Key Laboratory for Conservation and Utilization of Subtropical Bio-Resources, Education Department of Guangxi Zhuang Autonomous Region, Yulin Normal University, Yulin 537000, China
4   Crop Research Institute, Huzhou Agricultural Science and Technology Development Center, Huzhou 313000, China
5   Huzhou Key Laboratory for Innovation and Application of Agricultural Germplasm Resources, Huzhou 313000, China
*   Correspondence: 0617375@zju.edu.cn (X.Z.); dingyf@njau.edu.cn (Y.D.)

**Abstract:** Aluminum (Al) toxicity inhibits root growth, while nitrogen is an essential nutrient for plant growth and development. To explore the effects of nitrate (N) on Al toxicity and accumulation in root of wheat, two wheat genotypes, Shengxuan 6 hao (SX6, Al-tolerant genotype) and Zhenmai 168 (ZM168, Al-sensitive genotype), were used in a hydroponic experiment with four treatments (control without N or Al, N, Al, and Al+N, respectively). The results showed that N increased the inhibition of root elongation and aluminum accumulation in root. The Al-sensitive genotype suffered more serious Al toxicity than the Al-tolerant genotype. Histochemical observation clearly showed that Al prefers binding on the root apex 7–10 mm zones, and the Al-sensitive genotype accumulated more Al in these zones. Compared with other treatments, the Al+N treatment had significantly higher $O_2^-$, superoxides dismutase (SOD), catalase (CAT), peroxidase (POD) activities, $H_2O_2$, Evans blue uptake, malondialdehyde (MDA), ascorbic acid (AsA), pectin, and hemicellulose 1 (HC1) contents in both genotypes. Under Al+N treatment, $O_2^-$ activity, Evans blue uptake, MDA, and HC1 contents of SX6 were significantly lower than those of ZM168, but SOD, CAT, and POD activities and AsA content exhibited an opposite trend. Therefore, aluminum toxicity and accumulation in root of wheat seedlings were aggravated by nitrate.

**Keywords:** root elongation; aluminum toxicity; antioxidant enzyme; nitrate; wheat

## 1. Introduction

About half of the world and a quarter of China's cultivated land and potential cultivated land is characterized by acidic soil [1,2]. Unfortunately, more than 60% of acidic soils are located in developing countries, and these soils are critical for food production [3]. Aluminum (Al) is the most plentiful metallic element in the crust, usually existing in the form of non-toxic aluminosilicates and oxides in neutral soils. However, in acidic soils (PH < 5), rhizotoxic $Al^{3+}$ is solubilized into the soil solution and directly intoxicates root systems, which results in a significant reduction in crop yield worldwide [1,3]. The most typical symptom of Al toxicity is inhibition of root growth because Al mainly exists in root [4–6]. It has been well-documented that the root apex is not only the main site for Al perception and response, but also the target of Al accumulation [7–9]. The binding affinity to cell wall of root apex causes many adverse impacts, such as plasma membrane disagglomeration, signal disturbance, and reactive oxygen species (ROS) overproduction [10,11]. These disadvantages change the fraction of the cell wall and destroy its structure, thereby reducing its elasticity and plasticity, which explains the reason for inhibiting the elongation of root cells [12]. Moreover, Liu et al. (2018) [5] reported that Al-induced changes in ROS

are spatially specific, as a significant decreasing gradient is exhibited from the root apex to base.

Plants have evolved different strategies for coping with Al stress to maintain reasonable growth and yield [1]. One of the mechanisms of Al tolerance is the formation of a stabilized non-phytotoxic complex with Al by the secretion of organic acid anions from the root apex, thereby alleviating aluminum toxicity [13–15]. Another mechanism that endows Al tolerance is the enhancement of antioxidative defense capabilities [16]. Accumulating evidence supports that Al stress can alter the activity of enzymes associated with reactive oxygen species (ROS) scavenging [5,6,10].

Nitrogen is an essential nutrient for plant growth and development; the effects of nitrate on root development are well studied. Nitrate shows an inhibition effect on primary root growth but the opposite effect on lateral root [17,18]. Several vital genes involved in nitrate signaling pathways have been identified, including nitrate sensor, transcription factors, protein kinases, molecular components. Recently, Chu et al. (2021) [19] found a novel transcription factor (HBI1) that regulates nitrate signal transduction by mediating ROS homeostasis. They also found that nitrate treatment decreases the production of $H_2O_2$, and $H_2O_2$ inhibits nitrate signaling, thereby forming a feedback regulatory loop to regulate plant root development. A previous study also reported that nitrate can inhibit primary root growth by regulating the production of ROS in the root tips [17].

Al and N are important factors affecting root growth, finally impact crop yield. The uptake of nitrate accompanies the $OH^-$ secreting from roots, which increases the number of negative charge sites on the root surface for binding of $Al^{3+}$ [20] and simultaneously increases the pectin and hemicellulose owning to the negatively charged functional groups (e.g., COO- and -OH, respectively) that possess a high capacity for binding positively charged $Al^{3+}$ [21,22]. Root tips (0–10 mm) are generally used to investigate Al toxicity for the root growth of plants [22–24]. However, how N affects the Al toxicity and accumulation in root tips remain unclear. Thus, this study aimed to investigate the effects of N on Al toxicity in the root growth and Al accumulation in root tips of two wheat genotypes differing in Al tolerance by analyzing the root phenotype, histochemical staining, ROS, and antioxidant enzyme activity, as well as the cell-wall fractions in root tips.

## 2. Materials and Methods

### 2.1. Plant Materials and Treatments

Two wheat cultivars differing in Al tolerance, namely Shengxuan 6 hao (SX6, Al-tolerant) and Zhenmai 168 (ZM168, Al-sensitive), were used in this study. The seeds were soaked in distilled water for 1 h, and then disinfected with 1% NaClO solution by volume for 20 min and washed three times with deionized water to remove the residual NaClO on the seeds' surfaces, and then the seeds were imbibed for 12 h at 4 °C in refrigerated Petri dishes with filter papers in darkness. After the refrigeration, seeds were germinated at room temperature in darkness for 24 h. The germinated and uniform seeds were transferred to a plastic box containing 0.5 mmol/L $CaCl_2$ solution (pH 4.3). Wheat seedlings were incubated in an artificial climate chamber with a day/night cycle of 14 h/10 h, a temperature of 25 °C/20 °C, and a light intensity of 250 μmol photons $m^{-2}s^{-1}$. The solution was renewed daily.

After 4 days of pre-treatment, four treatments were adopted for 24 h, i.e., CK (0 mM $Ca(NO_3)_2$, 0 μM $AlCl_3$), N (5 mM $Ca(NO_3)_2$), Al (25 μM $AlCl_3$), and Al+N (25 μM $AlCl_3$ + 5 mM $Ca(NO_3)_2$). After 24 h, some seedings were used to determine the root length for calculating the relative root elongation and Al content. Parts of the samples' root tips (0–10 mm) were used for observing the traits by different staining methods. The rest of the samples' root tips were used to measure the antioxidant enzyme activity and cell-wall fractions.

### 2.2. Determination of Al Content of Cell Wall in Root Tips

The root tips (0–10 mm) were frozen at −80 °C for 12 h and then centrifuged to remove the cell saps; the residue was washed with 70% ethanol three times. The resulting cell-wall material was subsequently immersed in 0.5 mL 2 M HCl for 24 h with occasional vortexing. The Al content in the cell wall was determined according to Osawa and Matsumoto (2001) [25].

### 2.3. Localization of Al in Root Tips

The localization of Al was detected by hematoxylin and morin using the methods of Wu et al. (2020) [26]. Briefly, the treated root tips were soaked in 2 g/L hematoxylin with 0.2 g/L potassium iodide for 30 min. After washing for 30 min, the root tips were placed under a stereomicroscope MZ-95 (Carl Zeiss, Jena, Germany) for observation and photography. For morin staining, the root tips were immersed in 0.01% morin solution for 20 min, and then washed with deionized water for 10 min. Subsequently, root tip filming was taken by a laser scanning confocal microscope (Carl Zeiss LSCM 780, Jena, Germany) with a green fluorescence signal at 488 nm.

### 2.4. Membrane Integrity Verification Assay

The root tips were washed with deionized water three times for 5 min each time, stained in 0.25% ($w/v$) Evans blue solution for 15 min and rinsed three times with deionized water, and then observed and photographed under a visualization microscope [27]. Four stained root tips were weighed and milled in 1% sodium dodecyl sulfate (SDS) solution, centrifuged at 10,000 r/s for 10 min, the supernatant was determined at 600 nm, and the Evans blue uptake was calculated. The MDA content was measured by thiobarbituric acid (TBA) reaction according to Heath and Packer (1968) [28].

### 2.5. Antioxidant Enzyme Activity and Antioxidant Determination

Fresh roots were homogenized and extracted with 1 mL of 50 mmol/L sodium phosphate buffer (pH 7.0) containing 0.1 mol/L EDTA. After centrifugation, the supernatant was immediately used to determine the activities of antioxidant enzymes. Superoxide dismutase (SOD), peroxidase (POD), catalase (CAT), $O_2^-$ activities, $H_2O_2$, and ascorbic acid (AsA) contents were determined according to Liu et al. (2018) [5] with minor modification. All data were obtained by absorbance methods using a Tecan Infinite M200 Microplate Reader (Tecan Group Ltd., Männedorf, Switzerland). SOD activity was assayed by monitoring its inhibition of photochemical reduction of nitro blue tetrazolium (NBT) at 550 nm. POD activity was determined by following the change of absorption at 470 nm due to guaiacol oxidation. CAT activity was measured by following the consumption of $H_2O_2$ at 240 nm. $O_2^-$ activity, $H_2O_2$, and AsA were determined at 550 nm, 405 nm, and 536 nm, respectively.

### 2.6. Cell-Wall Fraction Determination

Extraction of cell-wall materials and the subsequent fractionation of cell-wall components were carried out according to Yang et al. (2011) [22] with minor modification. Fresh root tips were thoroughly homogenized with pre-cooled 75% ethanol and the homogenates were placed on ice for 20 min. Subsequently, they were centrifuged at 8000× $g$ for 10 min at 4 °C, and the residues were washed for 20 min in the order of acetone, a methanol:chloroform mixture (1:1, $v/v$), and methanol. The supernatant was discarded and the precipitates were freeze-dried.

The pectin fraction was extracted twice by 0.5% $(NH4)_2C_2O$ (ammonium oxalate) buffer containing 0.1% $NaBH_4$ (pH 4) in a boiling water bath for 1 h. The resulting residues were subsequently subjected to triple extractions with 4% KOH containing 0.1% $NaBH_4$ at room temperature for a total of 24 h, obtaining the hemicellulose 1 (HC1). The uronic acid content in each cell-wall fraction was calculated by a calibration standard curve generated with known concentrations of Galacturonic acid (GalA).

### 2.7. Statistical Analysis

The values in the figures were calculated as the mean ± SD. All data were analyzed using SPSS 17.0 software (Statistical Product and Service Solutions, IBM, Endicott, NY, USA). The statistical significance among treatments was determined through one-way ANOVA, followed by Duncan's multiple range test ($p < 0.05$), and significant differences were evaluated based on $p < 0.05$.

## 3. Results

### 3.1. Effect of N and Al on Root Elongation and Al Accumulation in Root Tips

An obvious inhibition of N and Al on root elongation was observed in both genotypes compared with CK (Figure 1A,B). For the relative root elongation, no significant difference was found between N and Al treatments in SX6, while significant differences were found in ZM168. Moreover, the relative root elongations of SX6 were markedly higher than those of ZM168 in the presence of Al (Figure 1A,B). A significantly higher Al content in the cell wall of root tips (0–10 mm) was found in both genotypes exposed to Al (Figure 1C). Furthermore, the Al content under Al+N treatment was much higher than that under Al treatment, while no difference of Al content was found under CK and N treatment. The Al content of ZM168 was significantly higher than that of SX6 when Al existed.

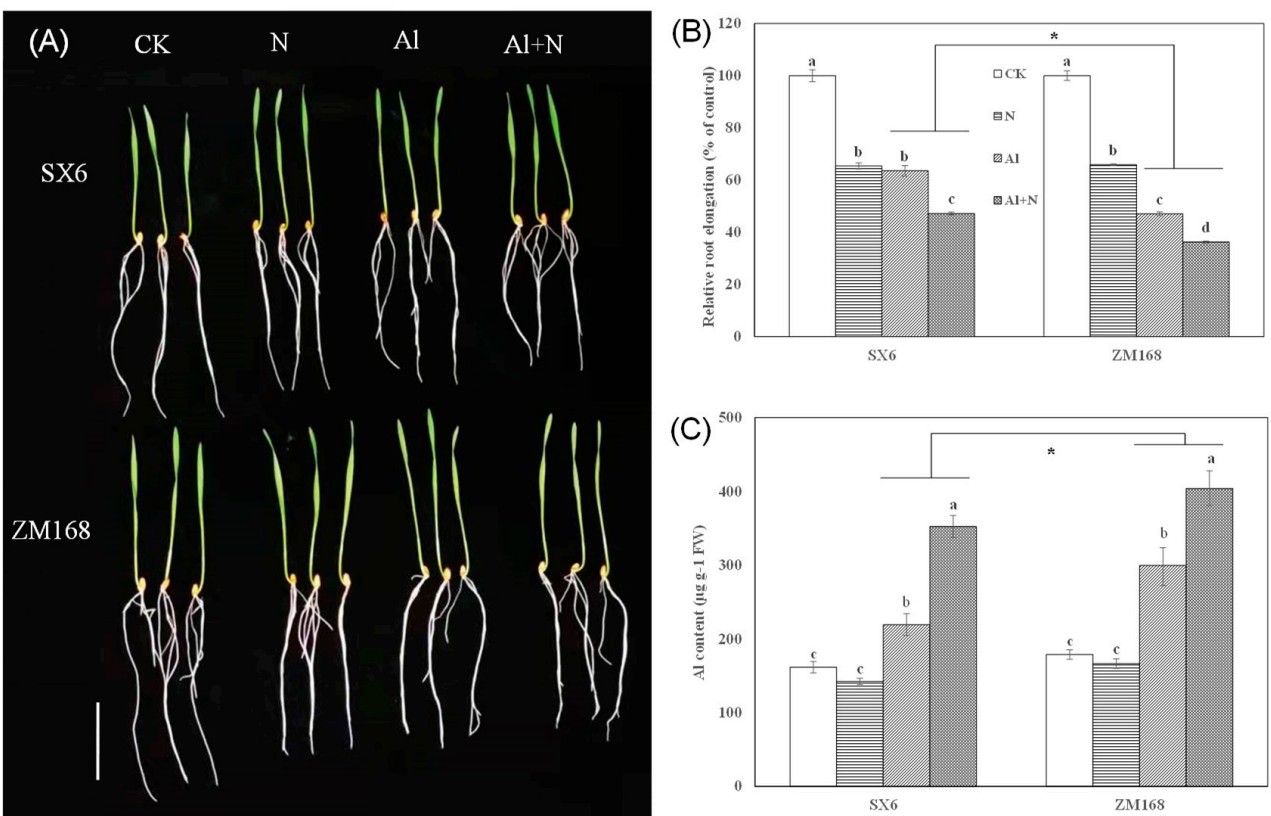

**Figure 1.** (**A**) Phenotypic analysis of Shengxuan 6 hao (SX6) and Zhenmai 168 (ZM168) seedlings in response to N, Al, and Al+N, scale bar = 5 cm. (**B**) Relative elongation was expressed relative to root elongation in control solutions of 0.5 mM CaCl$_2$, pH 4.3. (**C**) Al$^{3+}$ content of the cell wall in apical 0–10 mm root segments. The values shown are means ± SD ($n$ = 3). Different letters labeled on the columns in the same cultivar are significantly different ($p < 0.05$). **\*** stands for a significant difference in the same treatment between two cultivars ($p < 0.05$).

### 3.2. Al Localization in Root Tips

The Al and Al+N treatments significantly increased the Al accumulation obtained from hematoxylin staining (Figure 2A). It is noteworthy that more Al was accumulated in 7–10 mm and 0–3 mm zones under the Al+N treatment compared with Al treatment, especially for the Al-sensitive genotype (ZM168). To further verify these results, we used morin staining to examine the Al accumulation in those segments. As can be seen from Figure 2B, the root segments of both genotypes under Al+N treatment were brighter than those under Al treatment, exhibiting a synergistic effect, i.e., N promoted the accumulation of Al in the root tips (Figure 2). Al accumulated more in the 7–10 mm zones than in the 0–3 mm zones of root tip, and Al accumulation was higher in the Al-sensitive genotype (ZM168) than the Al-tolerant genotype (SX6) (Figure 2B).

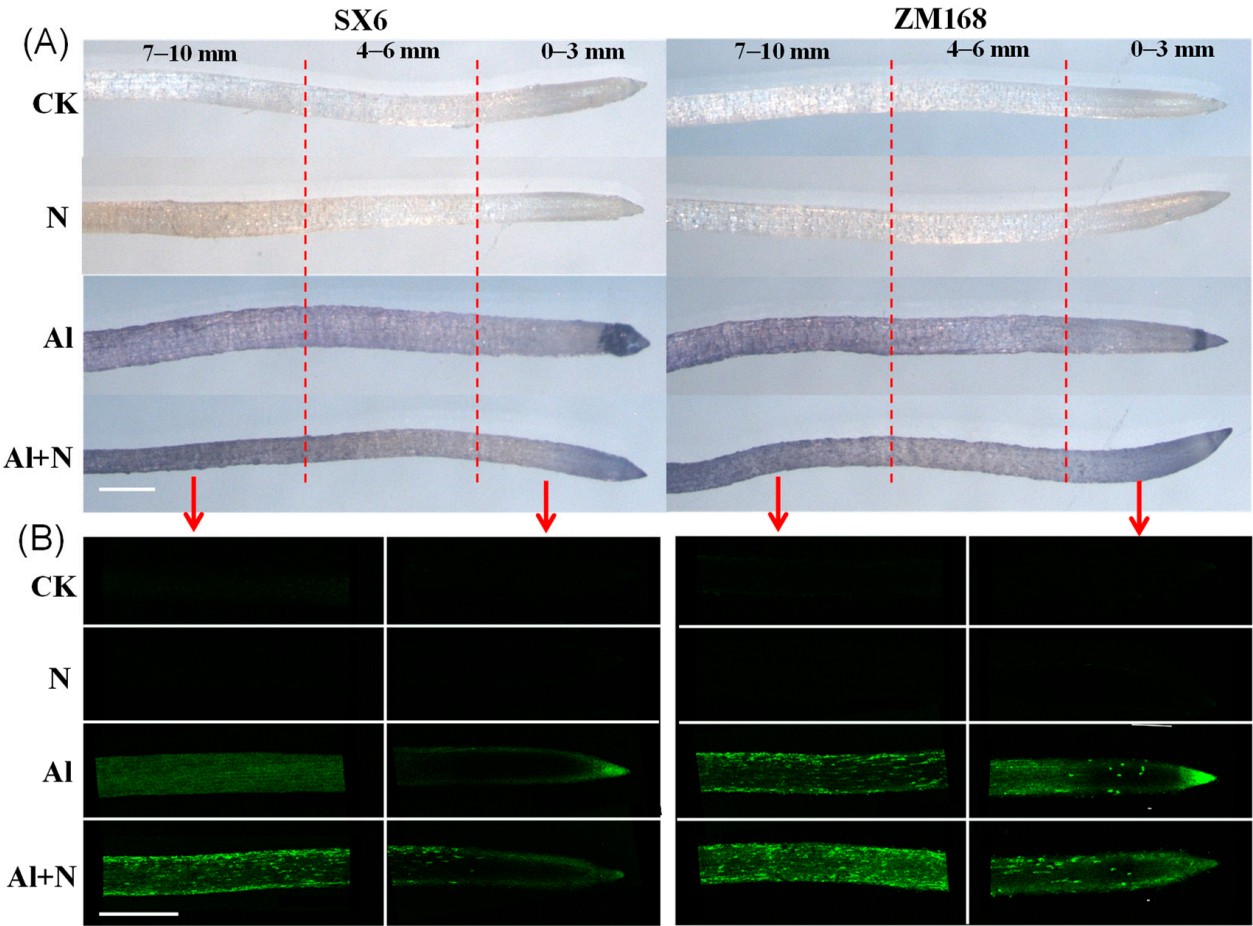

**Figure 2.** The Al localization ((**A**), hematoxylin staining and (**B**), morin staining) of root tips of two wheat genotypes (SX6 and ZM168) under CK, N, Al, and Al+N treatments. Scale bar, 1 mm.

### 3.3. Oxidative Damage and Reactive Oxygen Species (ROS) in Root Tips

The Evans blue staining observation showed that the darkest color was exhibited in Al+N treatment, followed by the Al treatment, then the N treatment, and the last being the control, which was confirmed by the relative value of Evans blue uptake (Figure 3A,B). Moreover, the picture showed that the cell wall of root 7–10 mm zones were the most seriously damaged in both genotypes under Al+N treatment (Figure 3A). The MDA content, $H_2O_2$ content, and $O_2^-$ activity in the root tips of SX6 and ZM168 showed the same pattern as the relative value of Evans blue uptake, except for the MDA and $H_2O_2$ content in SX16 under Al and N treatments (Figure 3C–E).

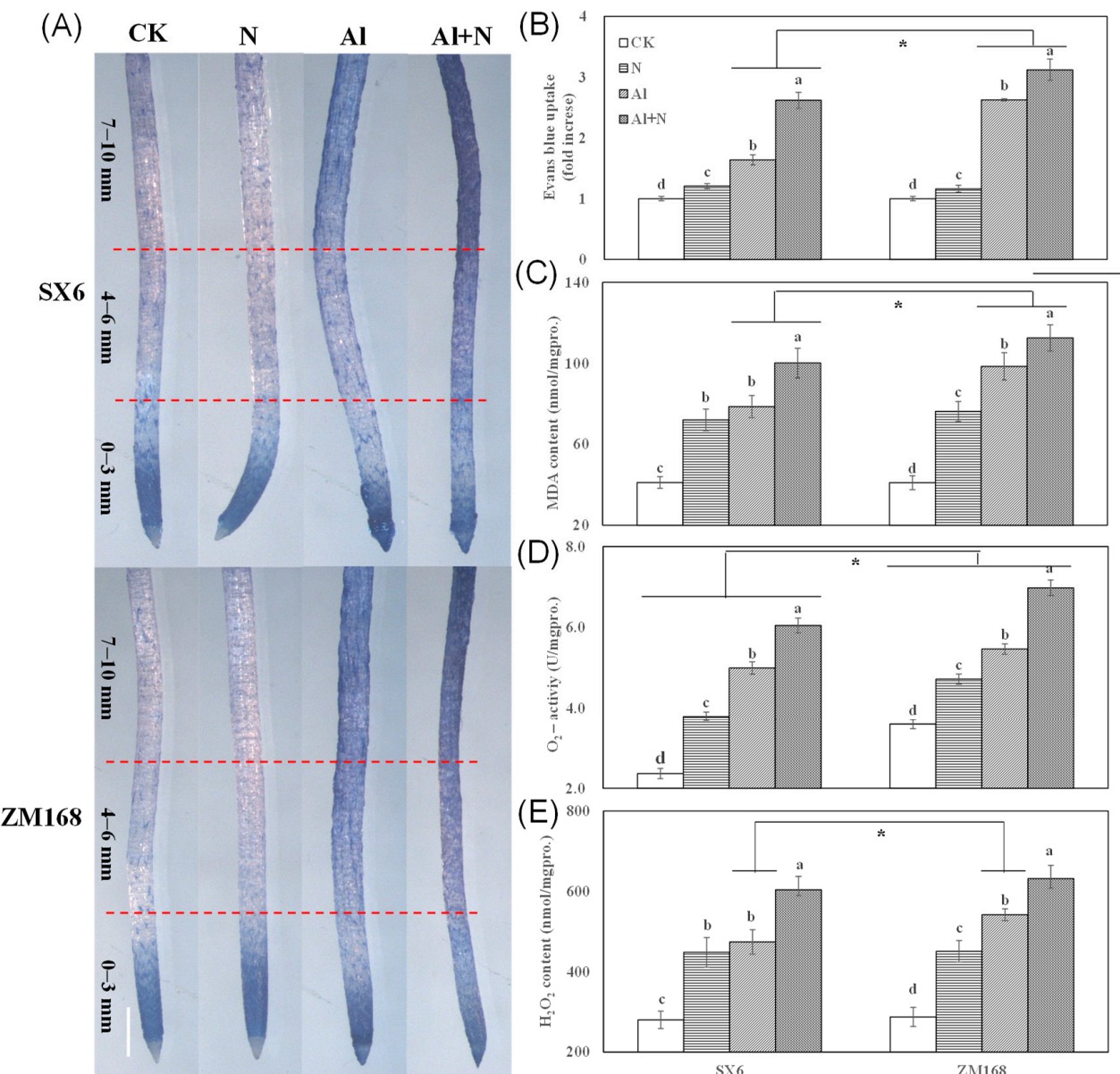

**Figure 3.** Oxidative damage parameters in the root tips of two wheat genotypes (SX6 and ZM168) when treated with different solutions. Root tips were collected 24 h after treatment, and then Evans blue staining (**A**), Evans blue uptake (**B**) (relative fold change was expressed compared with control), malondialdehyde (MDA) content (**C**), $O_2^-$ activity (**D**), and $H_2O_2$ content (**E**) were determined. The values shown are means $\pm$ SD ($n = 3$). Different letters labeled on the columns in the same cultivar are significantly different ($p < 0.05$). **\*** stands for a significant difference in the same treatment between two cultivars ($p < 0.05$). Scale bar, 1 mm.

### 3.4. Antioxidant Defense System and Cell-Wall Fractions

The activities of SOD, POD, and CAT in the root tips of both genotypes remarkably increased in the presence of Al compared with the control, and nitrate strengthened this trend. Compared with ZM168, SX6 had a higher SOD activity under Al+N treatment, higher POD activity under Al treatment, and higher CAT activity under all treatments (Figure 4A–C).

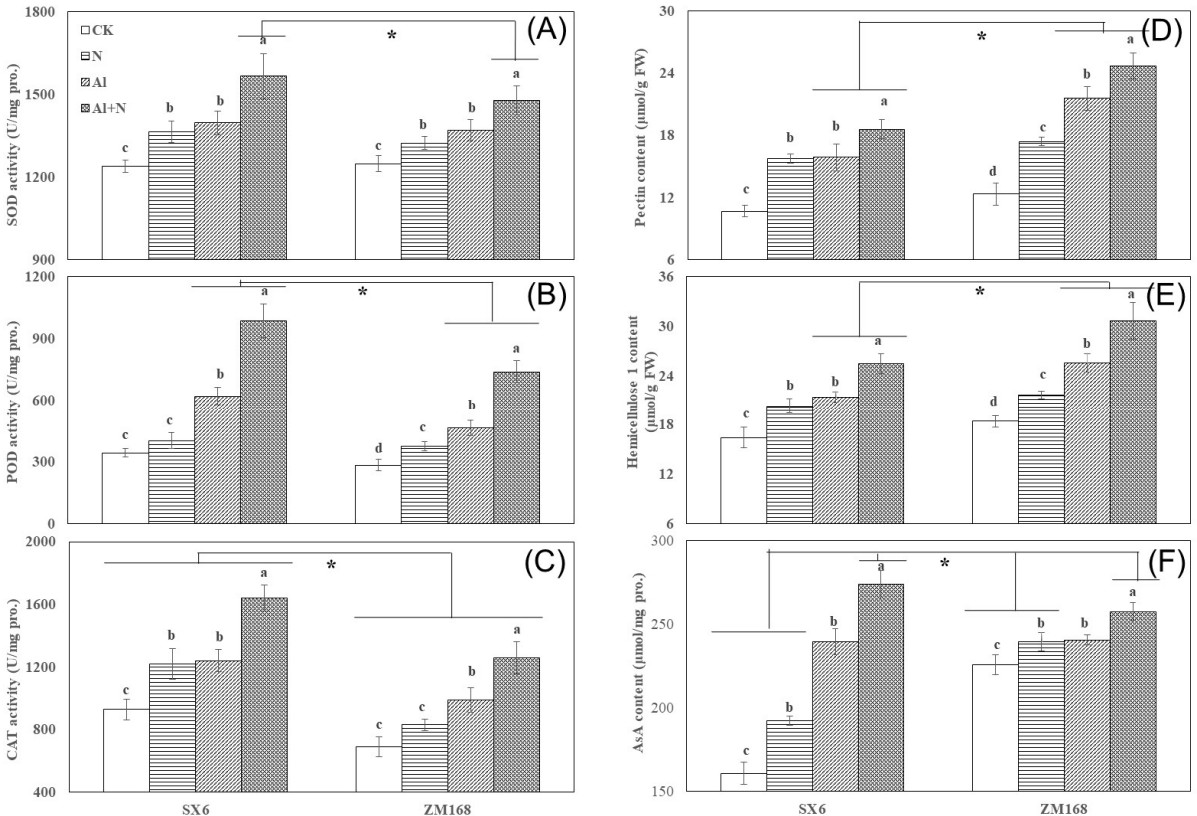

**Figure 4.** Effects of different treatments on antioxidant enzyme SOD (**A**), POD (**B**), CAT (**C**) activities and cell-wall fraction pectin content (**D**), HC1 content (**E**), and ascorbic acid content (**F**) in root tips. The values shown are means ± SD (*n* = 3). Different letters labeled on the columns in the same cultivar are significantly different ($p < 0.05$). * stands for a significant difference in the same treatment between two cultivars ($p < 0.05$).

There were significant differences on the variation amplitude of AsA content between two different genotypes. Compared with CK, the AsA content of SX6 rose steeply across treatments, increasing by 19.6%, 49.1%, and 70.6% under N, Al, and Al+N treatments, respectively, while the AsA content of ZM168 only increased by 6.2%, 6.5%, and 14.1% under N, Al, and Al+N treatments, respectively.

For the cell-wall components, compared with CK, the pectin content of SX6 increased by 48.7% and 73.9% under Al and Al+N treatments, respectively, while the pectin content of ZM168 increased by 74.8% and 100% under Al and Al+N treatments, respectively. A similar trend was found in the HC1 content, indicating that the cell-wall components changed more in the Al-sensitive genotype (ZM168) than in the Al-tolerant genotype (SX6) when root was exposed to Al or Al+N (Figure 4D,E).

## 4. Discussion

Root inhibition growth is a typical symptom of Al toxicity [6,29]. In this study, we found that the inhibition of root growth was the largest under Al+N treamtent in both genotypes, and nitrate promoted the accumulation of Al in root tips and an obvious synergistic inhibition occurred in the root elongation of both genotypes (Figure 1).

Moreover, the Al preferred accumulating at the root tip 0–3 mm and 7–10 mm zones, especially for the latter (Figure 2). Our findings are in close agreement with previous studies showing that Al accumulation was the highest at the 0–5 mm root apex in wheat [5] and the 0–3 mm root apex in buckwheat [30] containing the distal transition zone, which was the most Al-sensitive root apical region [9]. Though 0–2 mm zones in the root tip were considered as an indicator of genotypic sensitivity of crops to Al [9,30,31], the Al content in mature root 5–15 mm zones was four times higher than that in 0–2 mm zones under

25 μM $Al^{3+}$ concentration [32], which supports our result that 7–10 mm zones exhibited a higher Al deposition than that of 0–3 mm zones (Figure 2). Furthermore, nitrate promoted the accumulation of Al in the root tips 0–3 mm and 7–10 mm zones, and a larger impact was found in 7–10 mm zones, especially for the Al-sensitive genotype (Figure 2). Thus, we speculated that the deposition of Al in 7–10 mm zones was a main factor in limiting the root elongation and it could be promoted by N.

ROS production and conversion play an important role in root growth [33,34]. It is well documented that the accumulation of Al induces the formation of large amounts of ROS in crop roots, which damages cell membranes and may lead to cell death [35,36], and is a key factor in inhibiting root elongation [37]. In this study, the application of Al or N significantly induced the production of ROS, including $H_2O_2$ and $O_2^-$. Furthermore, the mixture of Al and N enlarged the production of ROS compared with single Al or N treatment (Figure 3D,E). Zang et al. (2020) [17] reported that nitrate inhibited the primary root growth by reducing the $H_2O_2$ content in *M. truncatula*, which was the opposite of our result, probably because the $H_2O_2$ had an opposite effect between *T. aestivum* [6] or *A. thaliana* [34] and *M. truncatula* [17]. To investigate the contribution of ROS in lipid peroxidation and cell viability, we also examined the MDA content and Evans blue uptake, and they exhibited similar patterns to $H_2O_2$ (Figure 3B,C,E). This result indicated that the massive production of $H_2O_2$ under Al stress may play a crucial role in the triggering of lipid peroxidation and cell death [35,36]. Thus, a lower ROS content in the Al-tolerant genotype (SX6) conferred a lower root growth inhibition compared with the Al-sensitive genotype (ZM168).

To alleviate the oxidative damage caused by ROS accumulation, plants have evolved a complex defensive antioxidant system that includes a combination of enzymatic and non-enzymatic components. Reactive oxygen enzymatic scavenging systems mainly include SOD, POD, and CAT, etc., and the activities of these enzymes respond to the strength of plant resistance in different degrees [35,38,39]. In plants, AsA is an important reductant and exerts a powerful influence on plant functions [40]. In the present study, the activities of SOD, POD, CAT, and AsA were significantly increased under N or Al treatments, except for CAT content in the Al-sensitive genotype ZM168 under N treatment. The highest values of antioxidant enzyme activities in both genotypes were found under Al+N treatment, and the values in ZM168 were significantly lower than those in SX6 (Figure 4A–C). Thus, roots of both genotypes suffered heavy oxidative damage to the membranes and lipids as a result of the higher level of ROS under Al+N treatment than that under single N or Al treatment, especially for the sensitive genotype ZM168.

The cell-wall polysaccharide fraction is considered as a novel Al resistance mechanism, since cell-wall binding capacity is related to Al accumulation in plant roots [22,41]. The cell wall is a major site of Al accumulation in crops, and more than 70% of Al binds to the cell wall of wheat root [22]. Al binding may disrupt the cell-wall structure and diminish mechanical extensibility, thereby inhibiting root elongation [24,42]. Pectin and HC1 are important components of the cell wall. In this study, Al exposure increased the contents of pectin and HC1, which appeared more prominently in the Al-sensitive genotype of wheat, consistent with a previous study [6]. Moreover, N significantly increased the contents of pectin and HC1 of the cell wall in the root tips for both genotypes exposed to Al+N (Figure 4E,F) compared with N or Al treatment, especially in ZM168, resulting in a higher Al accumulation in the cell wall of root tips (Figure 1C) under Al+N treatment. Therefore, N accelerated the accumulation of Al in the wheat root and enhanced the toxicity of Al.

## 5. Conclusions

Nitrate significantly increased Al accumulation in the roots of wheat seedlings and thereby intensified the inhibition of root elongation by Al. Al prefers to bind on the root apex 7–10 mm zones of the roots, and Al accumulation could be promoted by N. N increased reactive oxygen species (ROS), enzyme activities from the antioxidant defense system, and cell-wall polysaccharide fraction.

**Author Contributions:** Conceptualization, X.Z. and Y.D.; investigation, Y.M. and X.Z.; data analysis, Y.M.; writing—original draft preparation, Y.M. and C.B.; writing—review and editing, X.Z. and Y.D.; funding acquisition, X.Z. All authors have read and agreed to the published version of the manuscript.

**Funding:** This research was funded by National Natural Science Foundation of China (31901443).

**Institutional Review Board Statement:** Not applicable.

**Data Availability Statement:** Not applicable.

**Acknowledgments:** We are grateful to thank for kindly providing the wheat seeds by Huadun Wang, who comes from Jiangsu Academy of Agricultural Sciences. We also thank Qin Chen for providing the help in microscopic observation.

**Conflicts of Interest:** The authors declare no conflict of interest.

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
