# Peer review of "Nitrate Increases Aluminum Toxicity and Accumulation in Root of Wheat"

_agriculture, doi:10.3390/agriculture12111946_

Round 1

Reviewer 1 Report

This manuscript generally performed decent research with clear presentation about its results, but revisions are needed before it could be accepted. First, the reviewer suggests avoiding using ‘nitrogen’ throughout the manuscript; rather, ‘nitrate’ would be more accurate. After all, the authors did not use ammonium or organic nitrogen (e.g., urea) in the experiments, and ‘nitrate’ is not the sole form of nitrogen in soils/fertilizers. Second, some details need to be provided, including the rationale for the experimental design and further descriptions on the analytical methods (detailed questions and concerns can be referred in the following). Language-wise, some sentences are very long and hard to follow with occasional grammatical errors or redundancy. The reviewer would suggest the authors to find some native speakers to revise the language if possible. Advice has been given below for some obvious language issues:

Line 14: a conjunction word is needed between “Al toxicity inhibits root growth” and “N is an essential nutrient for plant growth and development”. Otherwise, split the sentence into two.

Line 18: should be “without N or Al”, not “N and Al”.

Line 21-24: In this sentence, there are too many “and”. You only need one “and”, which lies between your last and second last subject. Other than that, just use coma to separate the items.

Line 60: change “by mediated” into “by mediating”.

Line 70: The sentence that “Liu et al. (2018) found that the Al accumulation in root was spatial-dependent” has been similarly expressed in Line 46-47. It should be deleted here to avoid redundancy.

Line 86: How did the authors control the pH to maintain 4.3 throughout the experiments? Was buffer used or just using HCl to adjust the solution pH?

Line 90: What was the rationale to choose 5 mM to be the Ca(NO3)2 concentration and 25 uM to be the AlCl3 concentration? Do these values represent typical nitrate and aluminum ion concentrations in soil solutions?

Line 114: Please define SDS.

Line 121-125: It would be great if some published articles, books, online documents, or websites could be cited instead of just saying these enzyme activities were determined based on a company’s protocol. Future researchers may need detailed methods when they would like to do follow-up studies.

Line 170: Before “N promoted the accumulation …”, add “i.e.,” or “, that is,”

Line 196-198: This sentence needs to be re-written as it is too confusing. The authors made comparisons among genotypes and treatments at the same time. It is difficult to follow what are truly compared for.

Line 204-206: What is the comparison basis? The control?

Line 210, for Figure 4D and 4E, the Y axes are both uronic acid content. Is this a mistake? Based on the figure caption, 4D should be pectin content and 4E should be HC1 content.

Line 218-219: This sentence should be two separate ones. Otherwise, add an “and” before “N promoted…”

Line 223: change “where was” into “which was”.

Line 230: The sentence that “And N promoted…”is grammatically wrong since “and” cannot start a sentence. ‘Also’, ‘In addition’ or ‘Furthermore’ should be used instead.

Line 237: change ‘was a key factor’ into ‘is a key factor’. Please keep the verb tense consistent since ‘induces’ was used previously in this sentence.

Line 237-242: This sentence is way too long! Please split it into several sentences so that clear messages can be conveyed. Also, using two attributive clauses in a row is somewhat awkward (line 240-241).

Line 245-250: Again, such a long sentence with convoluted information conveyed. Please use more concise and effective expressions.

Line 254-256: Again, this contains two sentences. Add an “and” after “etc”.

Line 264-266: Delete everything starting from “underwent…” till the end of the sentence. These are unnecessary repetitions of the first half sentence.

Line 269: Add and “and” before “more than”, and change “was binding” into “is binding”.

Line 281: It is suggested to change this sentence into “Nitrate significantly increased Al accumulation in roots of wheat seedlings and thereby intensified inhibition of root elongation caused by Al.”

Line 283-286: This is a repetition of the abstract with the same grammar issue. Maybe just summarize it in this way: “N increased reactive oxygen species (ROS), enzyme activities from the antioxidant defense system and cell wall polysaccharide fraction” instead of listing every single item.

Author Response

Point 1: This manuscript generally performed decent research with clear presentation about its results, but revisions are needed before it could be accepted. First, the reviewer suggests avoiding using ‘nitrogen’ throughout the manuscript; rather, ‘nitrate’ would be more accurate. After all, the authors did not use ammonium or organic nitrogen (e.g., urea) in the experiments, and ‘nitrate’ is not the sole form of nitrogen in soils/fertilizers. Second, some details need to be provided, including the rationale for the experimental design and further descriptions on the analytical methods (detailed questions and concerns can be referred in the following). Language-wise, some sentences are very long and hard to follow with occasional grammatical errors or redundancy. The reviewer would suggest the authors to find some native speakers to revise the language if possible. Advice has been given below for some obvious language issues:

Response 1:

Thank you for your suggestion, we have changed ‘nitrogen’ to ‘nitrate’.

Point 2: Line 14: a conjunction word is needed between “Al toxicity inhibits root growth” and “N is an essential nutrient for plant growth and development”. Otherwise, split the sentence into two.

Response 2:

Thanks, we added “while” in the revised manuscript.

Point 3: Line 18: should be “without N or Al”, not “N and Al”.

Response 3:

We have changed it.

Point 4: Line 21-24: In this sentence, there are too many “and”. You only need one “and”, which lies between your last and second last subject. Other than that, just use coma to separate the items.

Response 4:

Thanks for your good suggestion. We have deleted the redundant “and” as you suggested.

Point 5: Line 60: change “by mediated” into “by mediating”.

Response 5:

It has been changed.

Point 6: Line 70: The sentence that “Liu et al. (2018) found that the Al accumulation in root was spatial-dependent” has been similarly expressed in Line 46-47. It should be deleted here to avoid redundancy.

Response 6:

It has been deleted.

Point 7: Line 86: How did the authors control the pH to maintain 4.3 throughout the experiments? Was buffer used or just using HCl to adjust the solution pH?

Response 7:

The solution was renewed daily to ensure a pH 4.3.

Point 8: Line 90: What was the rationale to choose 5 mM to be the Ca(NO3)2 concentration and 25 μM to be the AlCl3 concentration? Do these values represent typical nitrate and aluminum ion concentrations in soil solutions?

Response 8:

We chose 25 μM to be the AlCl3 concentration based on our previous experiment, 25 μM AlCl3 had a large inhibition on relative root elongation of both genotypes, and large differences between two genotypes were found under this AlCl3 concentration. 10 mM NO3- [5 mM Ca(NO3)2] concentration is considered as a NO3--rich condition, which begins to inhibit the root growth (Zhang and Forde, 1998).

Zhang, H.; Forde, B.G. An Arabidopsis MADS box gene that controls nutrient-induced changes in root architecture, Science 1998, 279,407-409.

Point 9: Line 114: Please define SDS.

Response 9:

We have defined the SDS.

Point 10: Line 121-125: It would be great if some published articles, books, online documents, or websites could be cited instead of just saying these enzyme activities were determined based on a company’s protocol. Future researchers may need detailed methods when they would like to do follow-up studies.

Response 10:

An article was cited and methods were described briefly.

Point 11: Line 170: Before “N promoted the accumulation …”, add “i.e.,” or “, that is,”

Response 11:

“i.e.,” was added.

Point 12: Line 196-198: This sentence needs to be re-written as it is too confusing. The authors made comparisons among genotypes and treatments at the same time. It is difficult to follow what are truly compared for.

Response 12:

We have re-written the sentence.

Point 13: Line 204-206: What is the comparison basis? The control?

Response 13:Yes, the comparison basis is control, “compared with CK” has been added.

Point 14: Line 210, for Figure 4D and 4E, the Y axes are both uronic acid content. Is this a mistake? Based on the figure caption, 4D should be pectin content and 4E should be HC1 content.

Response 14:

This is not a mistake, cell wall fractions pectin content (Figure 4D), HC1 content (Figure 4E) were all measured based on uronic acid content, but they were revised to avoid confusing.

Point 15: Line 218-219: This sentence should be two separate ones. Otherwise, add an “and” before “N promoted…”

Response 15:

We added “and” to this sentence.

Point 16: Line 223: change “where was” into “which was”.

Response 16:

We have changed it.

Point 17: Line 230: The sentence that “And N promoted…”is grammatically wrong since “and” cannot start a sentence. ‘Also’, ‘In addition’ or ‘Furthermore’ should be used instead.

Response 17:

The sentence that “And N promoted…” has the similar meaning with the former sentence, it was deleted to avoid redundancy.

Point 18: Line 237: change ‘was a key factor’ into ‘is a key factor’. Please keep the verb tense consistent since ‘induces’ was used previously in this sentence.

Response 18:

We have corrected it.

Point 19: Line 237-242: This sentence is way too long! Please split it into several sentences so that clear messages can be conveyed. Also, using two attributive clauses in a row is somewhat awkward (line 240-241).

Response 19:

We have re-written the sentence.

Point 20: Line 245-250: Again, such a long sentence with convoluted information conveyed. Please use more concise and effective expressions.

Response 20:

The sentence has been changed.

Point 21: Line 254-256: Again, this contains two sentences. Add an “and” after “etc”.

Response 21:

We added an “and” after “etc”.

Point 22: Line 264-266: Delete everything starting from “underwent…” till the end of the sentence. These are unnecessary repetitions of the first half sentence.

Response 22:

It has been deleted.

Point 23: Line 269: Add and “and” before “more than”, and change “was binding” into “is binding”.

Response 23:

We corrected them.

Point 24: Line 281: It is suggested to change this sentence into “Nitrate significantly increased Al accumulation in roots of wheat seedlings and thereby intensified inhibition of root elongation caused by Al.”

Response 24:

We have changed the sentence as you suggested.

Point 25: Line 283-286: This is a repetition of the abstract with the same grammar issue. Maybe just summarize it in this way: “N increased reactive oxygen species (ROS), enzyme activities from the antioxidant defense system and cell wall polysaccharide fraction” instead of listing every single item.

Response 25:

Thank you for your good suggestion, it has been changed.

Reviewer 2 Report

1. Found several statements about SX6 and ZM168 that are inconsistent with each other. For example, in the Abstract, it states "Shengxuan 6 (SX6, Al-sensitive genotype) and Zhenmai 168 (ZM168, Al-tolerant genotype)". In the main content (highlighted in yellow), it said: Al-sensitive genotype (ZM168) and Al-tolerant genotype (SX6)? In another place, it stated as "Al-insensitive genotype (ZM168)"?

2. Change all "u" to "µ" in the entire manuscript for the units µmol and µM.

3. Other suggestions/comments need to be addressed and revised (see the uploaded PDF file).

Author Response

Point 1: Found several statements about SX6 and ZM168 that are inconsistent with each other. For example, in the Abstract, it states "Shengxuan 6 (SX6, Al-sensitive genotype) and Zhenmai 168 (ZM168, Al-tolerant genotype)". In the main content (highlighted in yellow), it said: Al-sensitive genotype (ZM168) and Al-tolerant genotype (SX6)? In another place, it stated as "Al-insensitive genotype (ZM168)"?

Response 1:

It is a mistake in abstract, we have corrected it as "Shengxuan 6 hao (SX6, Al-tolerant genotype) and Zhenmai 168 (ZM168, Al-sensitive genotype)", and unified them throughout the revised manuscript.

Point 2: Change all "u" to "µ" in the entire manuscript for the units µmol and µM.

Response 2:

It has been changed.

Point 3: Other suggestions/comments need to be addressed and revised (see the uploaded PDF file).

Response 3:

Thank you for your good comments. We have revised according to your suggestions/comments in the PDF file.